# Experimental Evolution of Copper Resistance in *Escherichia coli* Produces Evolutionary Trade-Offs in the Antibiotics Chloramphenicol, Bacitracin, and Sulfonamide

**DOI:** 10.3390/antibiotics11060711

**Published:** 2022-05-25

**Authors:** Sada M. Boyd, Kristen L. Rhinehardt, Akamu J. Ewunkem, Scott H. Harrison, Misty D. Thomas, Joseph L. Graves

**Affiliations:** 1Department of Ecology and Evolutionary Biology, University of California, 612 Charles E. Young Drive South, Los Angeles, CA 90095, USA; 2Department of Computational Data Science and Engineering, North Carolina Agricultural and Technical State University, 1601 E. Market Street, Greensboro, NC 27411, USA; klrhineh@ncat.edu; 3Department of Biological Sciences, Winston Salem State University, 601 S. Martin Luther King Jr. Drive, Win-ston-Salem, NC 27110, USA; ewunkemaj@wssu.edu; 4Department of Biology, North Carolina Agricultural and Technical State University, 1601 E. Market Street, Greensboro, NC 27411, USA; scotth@ncat.edu (S.H.H.); mthomas1@ncat.edu (M.D.T.); gravesjl@ncat.edu (J.L.G.J.)

**Keywords:** *Escherichia coli*, copper, experimental evolution, genomics, antibiotics

## Abstract

The rise in antimicrobial resistant bacteria have prompted the need for antibiotic alternatives. To address this problem, significant attention has been given to the antimicrobial use and novel applications of copper. As novel applications of antimicrobial copper increase, it is important to investigate how bacteria may adapt to copper over time. Here, we used experimental evolution with re-sequencing (EER-seq) and RNA-sequencing to study the evolution of copper resistance in *Escherichia coli*. Subsequently, we tested whether copper resistance led to rifampicin, chloramphenicol, bacitracin, and/or sulfonamide resistance. Our results demonstrate that *E. coli* is capable of rapidly evolving resistance to CuSO_4_ after 37 days of selection. We also identified multiple de novo mutations and differential gene expression patterns associated with copper, most notably those mutations identified in the *cpx* gene. Furthermore, we found that the copper resistant bacteria had decreased sensitivity when compared to the ancestors in the presence of chloramphenicol, bacitracin, and sulfonamide. Our data suggest that the selection of copper resistance may inhibit growth in the antimicrobials tested, resulting in evolutionary trade-offs. The results of our study may have important implications as we consider the antimicrobial use of copper and how bacteria may respond to increased use over time.

## 1. Introduction

The emergence of multi-drug resistant (MDR) bacteria is a global public health concern, resulting in a need for antibiotic alternatives. One promising approach includes the use of metal-based antimicrobials that possess potent antibacterial abilities that are capable of killing a wide range of bacteria, including MDR strains [1,2]. The use of antimicrobial metals dates back thousands of years, although a general decline was observed when traditional antibiotics were introduced and started being used in the early 20th century [3]. However, the approach of a post-antibiotic era has prompted renewed interest in the use of antimicrobial metals.

The antimicrobial use of copper (Cu) and its alloys are of particular interest as a means of controlling bacterial infections within clinical and commercial settings. In February 2008, Cu became the first solid antimicrobial metal to be registered by the United States Environmental Protection Agency [4]. Generally, studies have indicated that the implementation of Cu surfaces in areas of healthcare facilities that are frequently touched, such as doorknobs, bed rails, and intravenous (IV) stands, reduces bacterial growth to levels that discourage nosocomial infections [5,6,7,8]. Consequently, the number of antimicrobial Cu-based products, along with studies investigating the mechanisms responsible for Cu’s bacterial-killing ability, have increased. In addition to an increased interest in antimicrobial copper surfaces, copper-dependent inhibitors (antibiotics that are only active in the presence of copper) are also gaining increased interest [9]. However, studies focusing on the potential of bacteria to evolve resistance to these novel uses of antimicrobial Cu over time are limited. That potential should be evaluated, due to the increase in antimicrobial Cu-based products.

Copper (Cu) is an essential micronutrient serving as a cofactor for redox-active cuproenzymes that are important for electron transfer and metabolic pathways [10] yet in excess, Cu is extremely toxic to bacterial cells [11]. Copper is commonly referred to in two forms, cupric Cu(I) and cuprous Cu(II). It is proposed that the bactericidal effects of Cu are attributed to the release of Cu ions, which removes electrons from the bacterial membrane, causing lipid peroxidation [12]. Additionally, C- induced damage is caused by Fenton-like chemical reactions that result in the generation of reactive oxygen species (ROS), causing DNA damage [13,14]. However, Macomber et al. (2007) showed that Cu toxicity does not contribute to oxidative DNA damage [15]. Furthermore, it is suggested that Cu ions can lead to the loss of membrane integrity [12] and destroy Fe-S clusters [16].

As a result of their multiple mechanisms of lethality, bacteria must carefully regulate Cu intracellular levels. To date, several regulatory pathways have been identified [17,18,19,20], including, the two-component response system, CusS/R, which induces the expression of the CusCFBA efflux pump that, in *Escherichia coli*, removes excess Cu(I) from the periplasmic space [18,19,20,21]. CueR is another transcriptional regulator that activates the transcription of *copA* and *cueO* [17,22,23,24]. The Cu translocating P-type ATPase, CopA, defends against the effects of excess Cu by transporting Cu from the cytoplasm to the periplasm [11,17,18], where the Cu efflux oxidase CueO is located. CueO provides further protection against Cu toxicity by reducing the more toxic form of Cu(I) to Cu (II) [23,24]; some gram-negative bacteria harbor specific (Cu^+^ oxidases and Cu^+^ chaperones) and non-specific outer membrane transporters participate in Cu detoxification and export [25].

Understanding the homeostatic and antibacterial mechanisms of any antimicrobial agent is important when accessing sustainability and there is no exception when it comes to copper. However, understanding the evolution of Cu resistance in bacteria is equally important, especially with the possibility of selecting for Cu resistant strains due to increased use. This understanding may help to provide the foundation needed to develop more sustainable Cu-based antimicrobials. Here, we use experimental evolution with genomic re-sequencing (EER-seq) and RNA-seq to investigate the genetics associated with the evolution of Cu (II) resistance in gram-negative bacteria using the model organism *Escherichia coli* K-12 MG1655. Additionally, we investigate whether Cu adaptation in *Escherichia coli* K-12 MG1655 leads to resistance or increased susceptibility in the presence of antibiotics.

## 2. Results

### 2.1. E. coli K-12 MG1655 Adapts to Excess CuSO_4_

After 37 days of selection in CuSO_4_, the Cu-selected populations displayed superior 24 h growth, according O.D._600nm_ measurements, in comparison to the control and ancestor populations in CuSO_4_ (Figure 1). Greater differences in growth were observed at 8–62 mg/L of CuSO_4_. There was a further difference between the ancestor and control populations, in that the control populations demonstrated slightly better growth in varying CuSO_4_ concentrations, but not population growth. F-statistics and their *p*-values for all comparisons are shown in Table 1.

To ensure that the effects observed in the Cu-selected populations were due to excess Cu and not to SO_4_, we grew all the populations in varying concentrations of MgSO_4_ (Figure 2)_._ We did not observe any differences between the ancestral, control, and Cu-selected populations in MgSO_4_.

### 2.2. Genomic Results

Whole-genome sequencing and analysis was conducted on all populations and compared to the *E. coli* K-12 MG1655 reference genome (NC_9000913) to determine genetic mutations that may be associated with Cu resistance. Genomic changes associated with the ancestor were reported in our previous studies and can be found in Appendix A, Table A1. Single nucleotide polymorphisms and selective sweeps were observed in the Cu-selected populations and reported in Table 2, along with the descriptions of the de novo mutations. We found six hard sweeps in the Cu-selected populations, with an increase in frequency from the ancestor (0.000) that exceeded 0.500 to 1.000 (fixation) in five genes, *yidL*, *yfcZ*, *rho*, *cpxP*, and *cpxA*. Of the hard sweeps, two were found in replicate Cu2, three were found in Cu4, one was found in Cu6 and one near-sweep was found in Cu8. Eleven *cpx* mutations were found in seven of the nine replicates, with seven found in *cpxA*, two found in *cpxR*, and two more found in *cpxP*, suggesting significance for adapted Cu resistance. We also identified two mutations in the *rpoD* gene, one at fixation and another major variant found only in Cu6. Three *rho* mutations (one hard selective sweep, one major variant, and another minor variant) were identified in Cu5, Cu6, and Cu9. Other hard sweeps were identified in Cu4, *yfcZ/fadL*, a UPF0381 family protein/long-chain fatty acid outer membrane transporter (UniProtKB—P10384), and *yidL*, an uncharacterized transcription regulator (UniProtKB—P31449). All other variants were found in the *adhE*, *yddA*, *yidK*, *yidL*, and *fis* genes.

Single nucleotide polymorphisms for the control populations are shown in the Appendix A, Table A2. One selective sweep was identified in the control populations: *atpG-F1* sector of membrane bound ATP synthase, gamma subunit (UniProtKB—P0ABA6). Multiple major and minor variants were also identified in *rpo* (*rpoA rpoB rpoC*)—DNA-directed RNA polymerase (UniProtKB—P0A7Z4) and *opp* (*oppB oppC oppF*)—oligopeptide transport and binding (UniProtKB—P77737). Other variants included *dgoR* (Galactonate operon transcriptional repressor: UniProtKB—P31460), *mutT*(8-oxo-dGTP diphosphatase: UniProtKB—P08337), *pfkA *(ATP-dependent 6-phosphofructokinase isozyme 1: UniProtKB—P0A796), and *malE* (Maltodextrin-binding protein: UniProtKB—P0AEX9) genes.

#### 2.2.1. Functional Annotation Analysis of Upregulated and Downregulated Genes

The DAVID functional annotation clustering tool grouped upregulated genes found in four or more Cu-selected replicate populations into one significant functional cluster—ABC transport and ATP binding—with a significant enrichment score of 1.3 (Table 3). Downregulated genes clustered into three significant functional clusters. The reported clusters were associated with bacterial flagellum, iron transport, and cell division (Table 3). The top two clusters each had an enrichment score of 1.3 or above and may be important for Cu resistance.

#### 2.2.2. Differences in Gene Expression for Upregulated and Downregulated Genes

Hierarchical cluster analysis of the differentially expressed genes was displayed using heatmaps that were generated using R version 3.5.3. The rows and columns of the heatmaps were clustered based on similar observations. Genes that were upregulated in at least four replicates with a fold-change of 2 or above that were not expressed in the controls are shown in Figure 3. Three genes involved in ABC (adenosine tri-phosphate binding cassette) transport, *mdtA*, *ybbP*, and *acrD*, were upregulated in at least half of the Cu-selected replicates. Both *acrD* and *ybbP* were upregulated in all eight of the replicates. AcrD is an aminoglycoside membrane efflux pump (UniProtKB—P24177); however, YbbP is uncharacterized. Another gene that was upregulated in all eight replicates was *xynR*, which encodes a transcriptional regulator of xylonate catabolism (UniProtKB—P77300). The most notable of the genes found in four replicate populations was *mdtA*, which encodes a multi-drug resistance-ABC transporter that requires the TolC efflux pump for function (UniProtKB—P76397). *moeA* involved in metal binding was upregulated in seven replicates. The most highly upregulated gene was *bioA*, adenosylmethionine-8-amino-7-oxononanoate aminotransferase, which is associated with biotin biosynthesis (UniProtKB—P12995). *bioP*, which is also responsible for biotin uptake (UniProtKB—P0ADP5), was upregulated in five replicates. The remaining genes, *yigA*, *hemF*, *spoT*, *uvrA*, and *ycjF*, were upregulated in at least four of the Cu-selected replicate populations.

There were many downregulated genes in four or more replicates (Figure 4). Most of the downregulated genes were associated with inner/outer membrane integrity, ion-transport, and flagella biosynthesis. Of the downregulated genes, *wbbL* was downregulated in all replicate populations. *wbbL*, which encodes for a rhamnosyltransferase, was involved in outer-membrane biogenesis (UniProtKB—P36667).

### 2.3. 24 h Growth Curves for Antibiotics

The 24 h growth measurements for each population (Cu-selected, control, and ancestor) were performed in varying concentrations of rifampicin, chloramphenicol, bacitracin, and sulfonamide to assess whether Cu resistance confers resistance to other antimicrobials (Figure 5A–D). F-statistics and *p*-values for each phenotype comparison are shown in Table 1. Generally, adaptation to Cu did not provide cross-resistance to the antibiotics tested in this study.

In the presence of chloramphenicol, the ancestors demonstrated slightly better growth when compared to the controls and the Cu-selected populations at 4–16 mg/L (Figure 5A). However, there was not a clear difference in growth between the controls and the Cu-selected populations in chloramphenicol (Figure 5A. In rifampicin, the controls demonstrated better growth when compared to the ancestor and Cu-selected populations (Figure 5B). The Cu-selected populations did, however, demonstrate slightly better growth compared to the ancestors. When comparing O.D. values in bacitracin, the ancestor and control populations showed superior growth compared to the Cu-selected populations in concentrations of 4–125 mg/L (Figure 5C). However, there was no difference observed in concentrations of 250 mg/L to 1000 mg/L. Similar results were found in the presence of sulfonamide, where the ancestors also had greater growth at mild concentrations (16–25 mg/L) when compared to the controls and the Cu-selected populations (Figure 5D). Generally, there was not a significant difference in growth when comparing the Cu-selected and control populations in the presence of sulfonamide.

## 3. Discussion

Little attention has been paid to the de novo evolution of copper resistance and the possible dissemination of copper-resistant bacteria. Even less attention has been paid to how evolved copper resistance may produce correlated resistances to other antimicrobials. Previous studies demonstrated that copper—in ionic and nanoparticles forms—is antibacterial, with MICs ranging from 32 µL/mL to 960 µg/mL [26,27,28]. Here, we showed that Cu resistance evolved rapidly in *E. coli* after 37 days of selection at a sub-inhibitory concentration of 75 mg/L CuSO_4_, via de novo mutations and gene expression changes. The Cu-selected populations grew better—i.e., they had higher O.D. values—than the control and ancestral populations at concentrations up to 62 mg/L of CuSO_4_ (Figure 1). Although the Cu-selected populations demonstrated superior growth from 2–62 mg/L, Cu selection did not improve growth in 125–1000 mg/L of CuSO_4_. This suggested that superior growth in Cu is limited to a selection concentration at or below 75 mg/L of CuSO_4_. However, the Cu-selected populations may be capable of adapting to higher concentrations of CuSO_4_ if this experiment were extended beyond 37 days of selection. In addition, we observed that the Cu-selected populations demonstrated superior growth in the absence of Cu, as shown in Figure 1. However, we did not explicitly investigate whether Cu selection leads to a selective advantage in a non-Cu environment, as that was beyond the scope of this study.

The most notable of our genomic results were the eleven mutations identified in *cpx*, which encompassed almost half of all the mutations identified. CpxAR makes up a two-component response system (TCRS) that responds to envelope stress [29], while CpxP inhibits the CpxAR response in the absence of that stress [30]. Previous studies have shown that CpxAR is activated in response to Cu toxicity and that excess Cu levels lead to the expression of genes regulated by CpxAR [22,29]. It has also been reported that CpxAR-controlled transcription may play a role in Cu tolerance in *Salmonella* by preventing Cu toxicity and restoring redox balance in the *Salmonella* envelope [31]. Additionally, adaptation to Cu stress in *E. coli* and *S. enterica* requires activation of the envelope stress response regulators [32,33], which is monitored by CpxAR [33,34]. The results of our study confirm that *cpxA* mutations are important for Cu adaptation caused by envelope stress stimulated by excess Cu [35]. This conclusion is supported by the fact that these mutations were only observed in the Cu-selected populations.

Populations with mutations in CpxR (PDB 4UHK) [36] include either an L20R mutations or a D104N mutation. As shown in Figure 6A, these two residues are located opposite one another at the ends of helices 1 and 5, respectively. Each of these mutations change the charge on the residue, likely disrupting helix stability within the structure of the protein. The Q128* mutation detected in CpxP results in a premature stop codon that would prevent translation of an entire helix (Figure 6B red). CpxP (PDV 3QZ3) [30] carries two conserved LTXXQ motifs (green) and Q128 (blue) in the second of these conserved motifs. Based on this mutation, which leads to a premature stop codon, it is very likely that this mutation disrupts the structure sufficiently so that it results in a non-functional protein. Finally, we were unable to map all CpxA (PDB 5LFK) [36] mutations onto structures available in the PDB, as only the cytoplasmic transmitter core domain was crystallized. Of the unmodeled mutations, three—R93H, V94A, and K121R—are located in the periplasmic sensor domain. Interestingly, previous work has shown that the deletion of residue 93–124 results in a constitutively active protein that upregulates the cpx regulons [37]. L178R is located within the second transmembrane helix and K188Q is the first residue of the HAMP domain directly out of the phospholipid bilayer. Unmodeled K194T and modeled P215L are also both located within the HAMP domain, and the modeled A262E mutation is located in the DHp domain [38]. These domains are important for structural rearrangement during activation and dimerization; all three mutations result in major changes in both size and charges of the residues, indicating that the function of this protein will likely be altered, although from current data we cannot yet predict whether that will be an increase or a decrease in activation/activity. Conversely, we identified multiple *rpo* mutations found in one Cu-selected replicate and in the control populations. We suggest that the *rpo* mutations found in our study may be important for adaptation to DMB, since *rpo* mutations were observed in the control populations of our previous studies [39,40,41,42,43].

Bacteria utilize metal transport systems to maintain metal homeostasis by allowing for the uptake of trace metals and the expulsion of excess metal, to avoid toxicity [44]. In bacteria, ABC transporters transport metal ions across the cell membrane to maintain metal homeostasis [45]. Among these transporters are P1B-type ATPases that export toxic levels of Cu from the cytosol [46]. It was predicted that the majority of transition metal transporting P1B-type ATPases responsible for cytoplasmic metal efflux are important for resistance to metal toxicity [46]. Thus, it was not surprising that a large percentage of the upregulated genes identified in multiple replicates were found in multiple ABC transporters pathways (Figure 3). In addition, according to our DAVID functional annotation cluster analysis, genes that fell within the category of ABC-transport/ATP-binding had the highest enrichment score (see Table 3). Two genes that were upregulated in multiple replicants—*acrD*, which encodes the ABC transporter aminoglycoside efflux pump, and *mdtD*, a multidrug efflux transporter gene—were shown to contribute to Cu resistance in *Salmonella* [47]. We suspect that the upregulation of *acrD* and *mdtD* may also contribute to Cu resistance in *E. coli*.

Surprisingly, *bioA* was upregulated in all eight Cu-selected replicate populations. The *bioA* gene encodes the protein adenosylmethionine-8-amino-7-oxononanoate aminotransferase, which catalyzes the transfer of the alpha-amino group from *S*-adenosyl-l-methionine (SAM) to 7-keto-8-aminopelargonic acid (KAPA) to form 7,8-diaminopelargonic acid (DAPA) (UniProt—P12995), which is a part of the biotin biosynthesis pathway. In biotin synthase, abstraction of a hydrogen atom from the C9 position of DTB leads to the formation of a dethiobiotinyl radical, which reacts with a nearby [2Fe-2S] 2 + cluster, generating a new carbon–sulfur bond [48]. It has been proposed that Cu toxicity causes damage to Fe-S clusters [16], perhaps upregulating *bioA*—a biotin synthesis pathway gene is a way that Cu-selected strains are accommodating for Cu ion-induced damage to biotin’s Fe-S clusters—but this proposal remains to be further elucidated. We also identified two additional genes associated with Fe-S clusters and biotin synthesis, *hemF* and *bioP*, to be upregulated in multiple replicates. 

We also identified many downregulated genes in our gene expression studies. Based on our DAVID functional annotation clustering analysis for downregulated genes, the top cluster was associated with bacterial flagella structural components and synthesis (see Table 3). Flagellum is a motility organelle that participates in protein export, adhesion, and biofilm formation. In *E. coli*, the regulation of flagella genes is important for the transition from the planktonic state to the biofilm state [49]. Although flagella are needed for the initial formation of biofilms, previous studies suggested that the inhibition of motility may promote biofilm formation [49]. Flagella synthesis genes *flgM *(UniProt—P0AEM4), a negative regulator of flagellin synthesis, and *flgN *(UniProt—P43533, an initiator of filament assembly biosynthesis, were downregulated in multiple Cu-selected populations. In addition, structural components flagella-basal body genes, *flgB* and *flgG* were downregulated for multiple replicates (UniProt-P0ABW9, UniProt-P0ABX5). It is well known that biofilms are inherently more resistant to metals than planktonic cells. Perhaps downregulation of flagella genes may be important for Cu adaptation.

The genes *waaO* and *waaL* both of which serve important roles in lipopolysaccharide synthesis, were downregulated in all eight replicates. Lipopolysaccharides are molecules located in the outer membrane of Gram-negative bacteria, which are important for the structural integrity of the bacterial cell. WaaL(rfaL) is involved in the LPS core biosynthesis pathway, which is part of the bacterial outer membrane biogenesis (UniProtKB—P27243). The WaaL protein adds the O-antigen on the glucose group of LPS. The O antigen provides protection from the antimicrobial action of certain antibiotics. WaaO (rfaI) is an LPS 1,3 galactosyltranferase involved in the synthesis of the R core lipopolysaccharide (UniProtKB—Q9ZIS5). Cytoplasmic stress response is activated upon the block of LPS, under the control of the stress response system CpxAR, which regulates the expression of genes involved in outer membrane biogenesis. Although not previously described, we suggest that the downregulation of *waaL* and *waaO* may be important for Cu resistance. Although the results of our gene expression studies are interesting, future studies are needed to better assess our hypothesis, and proteomic studies may provide further insight into how these genomic changes impact Cu resistance.

The potential cross-resistance across different antimicrobials, such as metal antibiotic antimicrobials, may hinder long-term use of antimicrobial metals. To test this, the control, ancestor, and Cu-selected bacteria were grown in various concentrations of antibiotics. Generally, Cu-adapted populations grew worse in the presence of the antibiotics, chloramphenicol, bacitracin, and sulfonamide, when compared to the control and ancestor populations (Figure 5), resulting in evolutionary trade-offs. However, our findings contrast with previous reports [50,51,52]. For example, Yu et al. (2022) found that Cu hydroxide-resistant *E. coli* were also resistant to chloramphenicol and tetracycline [50]. Other studies have shown that Cu exposure co- or cross-selects for resistance to antibiotics [53,54]. Generally, those studies isolated bacteria from Cu-contaminated environmental sites and attributed copper and antibiotic resistance to exogenous genes [50,51,52,53,54]. Here, we selected for Cu resistance using a laboratory strain without plasmids; accordingly, all Cu-associated mutations and correlated responses were the result of changes found on the chromosome. Additionally, the results of this study may have important implications for infectious disease control. For example, previous studies on antimicrobial copper primarily focused on the effect of incorporating copper surfaces in commonly touched areas in clinical settings. Generally, these studies indicated that the incorporation of copper-based surfaces reduces bacterial growth to levels that discourage nosocomial infections in healthcare facilities [5,6,7,8,55,56,57]. However, research that focused on the potential of bacteria to evolve resistance to antimicrobial copper in clinical settings over time is limited. Results indicate that Cu-resistance occurs rapidly and should be further explored. We also showed that Cu-resistance increased chloramphenicol, bacitracin, and sulfonamide sensitivity, suggesting that copper may provide a nonpharmacological approach to increasing antibiotic susceptibly. However, further studies are needed to investigate this hypothesis.

Although *cpx* mutations have been associated with antibiotic resistance in *E. coli* [58], we suspect that metal-antibiotic cross-resistance was not observed, as *cpx* mutations found in our study may impact downstream processes that are associated with copper homeostasis. However, further investigations are needed to determine the validity of this hypothesis.

Interestingly, the control populations showed superior growth in the presence of rifampicin. We suspect that *rpo* mutations resulting from adaptation to the DMB media may have contributed to this result.

As antibiotic resistance continues to rise, it is important to understand how bacteria may adapt to antibiotic alternatives. As a result, understanding how background resistances impact bacteria’s ability to develop future resistance may provide a greater fight in creating sustainable antimicrobials. Overall, the results of our study demonstrate that *E. coli* rapidly evolves resistance to excess Cu via genomic changes, and that Cu adaptation limits its ability to persist in the presence of the other antimicrobials tested. These results may have important implications for the use of Cu as an antimicrobial.

## 4. Materials and Methods

### 4.1. Bacterial Strains and Culture Media

*E. coli* K-12 MG1655 (ATCC #47076) was used for this study. It was chosen because of its sensitivity to metals and its lack of metal or antibiotic resistance. *E. coli* K-12 MG1655 does not contain any plasmids and its chromosome has 4,641,652 nucleotides (GenBank:NC_000913.3). All growth experiments in this study were carried out using Davis Minimal Broth (DMB, Difco^TM^, Sparks, MD, USA) supplemented with 1 g per liter dextrose as a sole carbon source, with thiamine hydrochloride 0.1% (Thiamine Hydrochloride, Fisher Scientific, Fair Lawn, NJ, USA).

### 4.2. Experimental Evolution

The experimental evolution by serial transfer method used in this study was modeled after our previous studies [39,40,41,42,43]. To start the evolution experiment, the *E. coli* K-12 MG1655 stock strain was cultured in 10 mL of DMB, serial-diluted, and plated onto DMB agar plates. Four individual colonies were picked and grown overnight in standard DMB media. To determine the minimum inhibitory concentration (MIC), overnight cultures were diluted to 0.05 O.D._600nm_ and added to 96-well plates in triplicate in 2× fold increasing concentrations of CuSO_4_ (0–5000 mg/L) diluted in autoclaved H_2_O. O.D._600nm_ measurements were taken at 0 and 24 h. The 0-h O.D._600nm_ measurement readings were subtracted from the 24 h readings and averaged for statistical analysis. An aliquot of the original culture was stored at −80 °C in 50% glycerol to be used as the ancestral strain. Once the MIC was determined, a sub-inhibitory concentration of CuSO_4_ (II) (75 mg/L) was used to start the selection experiment. Ten CuSO_4_ selected (Cu-selected) populations and five control populations originated from independent colonies from the same culture as the *E. coli* K-12 MG1655 ancestor. Each population was maintained in 50 mL Erlenmeyer flasks (Pyrex, Corning New York, NY, USA) that contained 10 mL (0.1 mL of each culture into 9.9 mL of DMB) of total culture were placed in a shaking incubator at 250 rpm (I2500 series, New Brunswick Eppendorf, Hamburg, Germany), at 37 °C for 24 h. Cultures were propagated by daily transfers. The selection experiments were continued until Cu resistance was confirmed by phenotypic assays (37 days of selection). The Cu-selected replicate populations were maintained with the addition of 75 mg/L of CuSO_4_, while controls were maintained in standard DMB only. Aliquots of each population were stored at −80 °C with 50% glycerol once a week to be used as a “fossil record”. There was a total of ten Cu-selected replicate populations; however, one replicate went extinct at approximately 14 days, leaving a total of nine replicates (Cu1, Cu2, Cu3, Cu4, Cu5, Cu6, Cu8, Cu9, and Cu10) and five controls (C1, C2, C3, C4, and C5).

### 4.3. 24 h Growth Assay in CuSO_4_

All the phenotypic growth assays used archived glycerol stocks that were grown overnight in 10 mL DMB supplemented with 75 mg/L of CuSO_4_ for the Cu-selected populations. Overnight cultures were diluted to optical density (O.D.) measurements at 600 nm (O.D._600nm_) of 0.05 and added to 96-well plates in triplicates. Phenotypic assays were performed by taking O.D._600nm_ at 0 and 24 h for the ancestor, control, and Cu-selected populations in 0, 2, 4, 8, 31, 62, 125, 250, 500, and 1000 mg/L of CuSO_4_ diluted in autoclaved H_2_O. The 0 h O.D._600nm_ measurement readings were subtracted from the 24 h readings and averaged for statistical analysis. Cu-selected replicate populations were deemed resistant when Cu-selected replicates demonstrated superior growth—i.e., higher O.D._600nm_ measurements—in varying CuSO_4_ concentrations when compared to the control and ancestor populations.

### 4.4. 24-h Growth Assay in MgSO_4_

To ensure that the Cu-selected populations evolved resistance to Cu, and not SO_4_ ions, we measured the 24 h growth of the ancestor, control, and Cu-selected populations in MgSO_4_. Archived glycerol stocks were grown overnight in 10 mL DMB supplemented with 75 mg/L of CuSO_4_ for the Cu-selected populations. Overnight cultures were diluted to O.D._600nm_ of 0.05 and added to 96-well plates in triplicate. Phenotypic assays were performed by taking O.D._600nm_ at 0 and 24 h for the ancestor, control, and Cu-selected populations in 0, 2, 4, 8, 31, 62, 125, 250, 500, and 1000 mg/L of MgSO_4_ diluted in autoclaved H_2_O. The 0 h O.D._600nm_ measurement readings were subtracted from the 24 h readings and averaged for statistical analysis.

### 4.5. Whole-Genome Sequencing and Sequence Analysis

To identify single nucleotide polymorphisms and insertion element localization associated with Cu resistance in *E. coli* K-12 MG1655, we employed whole-genome resequencing after resistance was observed at 37 days of selection. DNA samples from the Cu-selected, control, and ancestral populations were extracted and purified via the Nucleic Acid Sample Preparation for Downstream Analysis: Principles and Methods Kit (GE Health Sciences 2014).

Genomic libraries were prepared using the following standard protocol, as described in the manufacturer’s instructions (Illumina, Nextera DNA Flex Library Prep Reference Guide Part # 20,018,704, 20,018,705). The genomic libraries were sequenced using the Illumina MiSeq platforms at the Joint School of Nanoscience and Nanoengineering (JSNN). The depth of sequencing was between 40–80× for all samples. Ancestral mutations associated with our sample of this strain were removed from the analysis and the reads were mapped to the reference genome for *E. coli* K-12 MG1655 (NC_000913). Sequence alignment and variant calling from the samples was achieved by use of the *breseq* 0.30.0 pipeline [59]. The *breseq* pipeline uses three types of evidence to predict mutations, which include read alignments (RA), missing coverage (MC), and new junctions (JC). Any reads that indicate a difference between the sample and the reference genome that cannot be resolved to describe precise genetic changes are listed as “unassigned” [59]. These unassigned reads were not described or interpreted here. This analysis was used to identify genomic differences between control and Cu-selected bacteria, suggesting which physiological systems may be contributing to Cu resistance.

### 4.6. RNA Extraction

Once adaptation was observed at 37 days of CuSO_4_ selection, eight Cu-selected replicate populations (Cu1, Cu3, Cu4, Cu5, Cu6, Cu8, and Cu9) and four controls (C1, C2, C3, and C5) were prepared for RNA-sequencing. Cu4 and C1 were omitted due to low RNA quality. The gene expression data of the ancestral population is not shown, as it is not adapted to Cu or to the Davis Minimum Broth (DMB). Each replicate population was cultured under two conditions, Davis Minimum Broth (DMB) only and DMB with a sub-inhibitory concentration (75 mg/L) of CuSO_4_ in a shaking incubator at 37 °C at 250 rpm. A total of 1 × 10^6^ cells were harvested in mid-log phase at an O.D._600_ of 0.50. Total RNA was extracted, according to the manufacturer’s instructions, using the Monarch Total RNA Miniprep Kit. We further evaluated the RNA samples with a QuantiFluor RNA system, using the manufacturer’s standard protocol. Total RNA was stored at −80 °C prior to shipping to Novogene Corporation Inc. (Sacramento, CA, USA) for library construction and RNA-sequencing.

RNA degradation and contamination were monitored on 1% agarose gels. The RNA concentration and integrity number (RIN) were measured on an Agilent Bio Analyzer 2100 system. Only RNA samples with a RIN > 6 were used for cDNA library construction. Therefore, eight Cu-selected and four controls were cultured in a sub-inhibitory concentration of CuSO_4_ and four Cu-selected and one control cultured in DMB were used in this experiment. All cDNA libraries were sequenced using a paired-end strategy (read length 250–300 bp) on an Illumina NovaSeq 6000 platform. Novogene stringently analyzed the sequencing data to ensure accuracy and reliability, including, quality, error rate, Q20, Q30, and adapter contamination rate. Raw reads with low quality, containing adapter sequences, or with >10% of “N” base calls were removed to obtain the clean reads.

### 4.7. Differential Gene Expression Analysis

To analyze gene count data, the STAR output was input into DESeq2 R/Bioconductor package version 1.27.27, with the aim of finding genes that are differentially expressed across populations for the two conditions, DMB only and DMB with CuSO_4_. DEseq2 uses a negative binomial generalized linear model to analyze high-throughput sequencing data [54]. DESeq2 incorporates count data as a two-column table; for each sample, the number of sequence fragments that have been assigned to each gene [60]. The count data table was generated from the STAR ReadsPerGene.out.tab file, where data in the first and the fourth columns were extracted shown in the Appendix A, Table A3.

Each bioconductor software package often has a special class of data object, which contains special slots and requirements [60]. In the case of DESeq2, the data object is DESeqDataSet, which was used to store read counts and the intermediate estimated quantities during statistical analysis. The simplest design formula for differential expression would be ~condition, where each condition is a column in colData (dds) that specifies which group the samples belong to [60]. This was used to identify differentially expressed genes across populations for the two conditions, DMB (Davis Minimal Broth) only and DMB with CuSO_4_.

#### 4.7.1. Fold-Change Analysis

DESeq2 was used to calculate the fold-change differences for four pairwise comparisons, as shown in Appendix A, Table A4. Group 1 compares the Cu-selected populations in CuSO_4_ to the control populations in CuSO_4_. Group 2 compares the Cu-selected populations in CuSO_4_ to the Cu-selected populations in DMB only. Group 3 compares the control populations in CuSO_4_ to the control populations in DMB only. Group 4 identified uniquely expressed genes in the Cu-selected populations in CuSO_4_ that were not expressed in the controls in CuSO_4_ when compared to both replicate populations in DMB. Essentially, Group 4 was the result of comparing Groups 1 and 3 and was used for all the further analysis in this study (Appendix A, Table A3).

#### 4.7.2. Functional Annotation of Expressed Genes

The Database for Annotation, Visualization, and Integrated Discovery (DAVID), version 6.8, was used to group differences in expressed genes based on functional similarity to enhance biological interpretation. DAVID is comprised of multiple functional annotation tools used to cluster large sets of genes to understand the biological meaning behind large lists of genes. Here, we used the functional annotation clustering tool to classify uniquely upregulated and downregulated genes associated with Cu resistance. This tool suite provides typical batch annotation and gene-GO term enrichment analysis to highlight the most relevant GO terms associated with a given gene list [61]. Genes were considered upregulated if the fol-change was at 2 or above and downregulated if fold-change was at 0.5 and below. Functional annotation clustering integrates the same techniques of kappa statistics to measure the degree of the common genes between two annotations, to classify the groups of similar annotations according to kappa values [61]. Thus, genes with common annotation themes will most likely be grouped together. The functional annotation tool identified four functional clusters for upregulated genes based on the enrichment score (see Table 3). The enrichment score ranks the overall importance of annotation term groups. It is the geometric mean of all the enrichment P-values (EASE scores) of each annotation term in the group [61]. The geometric mean is a relative score instead of an absolute P-value, minus log transformation, which is applied on the average P-values. An enrichment score of 1.3 would be the equivalent to a non-log score of 0.05. Therefore, a higher score for a group indicates that annotation term members in the group are playing a more important (enriched) role in a given study [61].

#### 4.7.3. Differential Gene Expression

Hierarchical cluster analysis of the differentially expressed genes was displayed using heatmaps that were generated using R version 3.5.3. Genes that were upregulated in at least four replicates with a fold-change of 2 and above were considered upregulated, while genes 0.5 or below found in at four least replicates were deemed downregulated. We chose genes that were upregulated or downregulated in at least four replicates to represent the expression of genes found in at least half of the Cu-selected populations.

### 4.8. 24-h Growth Assay Antibiotics

To determine whether adaptations to excess CuSO_4_ (CuII) lead to cross-resistance in other antimicrobials, we measured the growth of the ancestor, control, and Cu-selected populations in varying concentrations of rifampicin, chloramphenicol, bacitracin, and sulfonamide. Optical density (O.D.) measurements at 600 nm were taken at 0 and 24 h in 0, 4, 8, 16, 31, 62, 125, 250, 500, and 1000 mg/L for each antimicrobial. Archived glycerol stocks for each population were grown and diluted to O.D._600nm_ of 0.05 and added to 96-well plates in triplicate, as described in the previous section.

### 4.9. Statistical Analysis

All statistical analysis comparing the 24 h growth rate for the ancestor, control, and Cu-selected populations in Cu and antibiotics were conducted using an ANOVA via the General Linear Method option on IBM SPSS ver. 23. ANOVA was used to determine the effect of population type (Cu-selected, control, ancestor), the concentration, and the interaction of population type and concentration on 24 h growth in varying concentrations of the metals and antibiotics tested.

## 5. Conclusions

The widespread use of antimicrobial copper may contribute to the emergence of copper-resistant bacteria. Furthermore, potential cross-resistance across antimicrobials, such as metal-antibiotic resistance, may hinder the long-term use of metals as antimicrobials. The results of this study indicate that copper resistance in *E. coli* can occur rapidly—i.e., in 37 days—via genomic changes reducing the ability to grow in the presence of the antibiotics tested. Our results may have important implications for the continued and future use of antimicrobial copper.

## Figures and Tables

**Figure 1 antibiotics-11-00711-f001:**
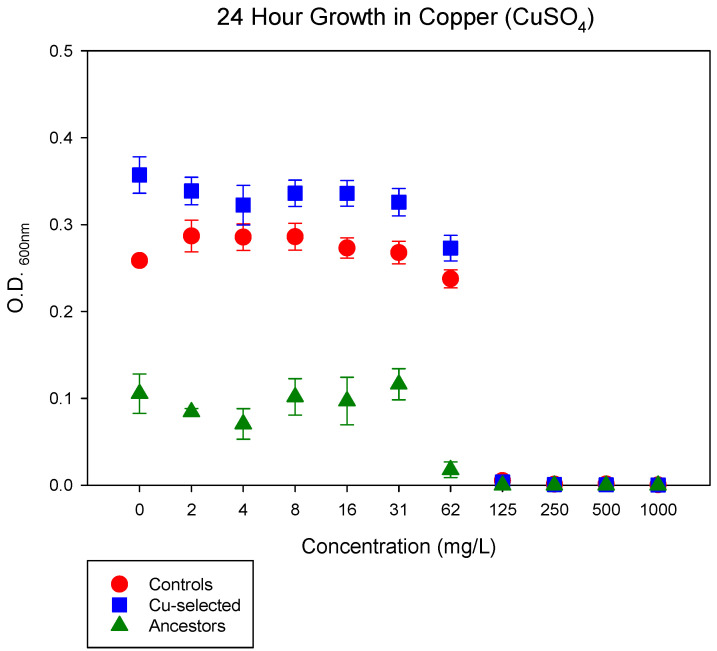
The mean and standard error (SE) of 24 h growth for Cu-selected, control, and ancestor populations in increasing concentrations of CuSO_4_ are shown. Cu-selected replicate populations demonstrated superior growth in CuSO_4_ concentrations of 8 mg/L to 62 mg/L when compared to the control and ancestor populations.

**Figure 2 antibiotics-11-00711-f002:**
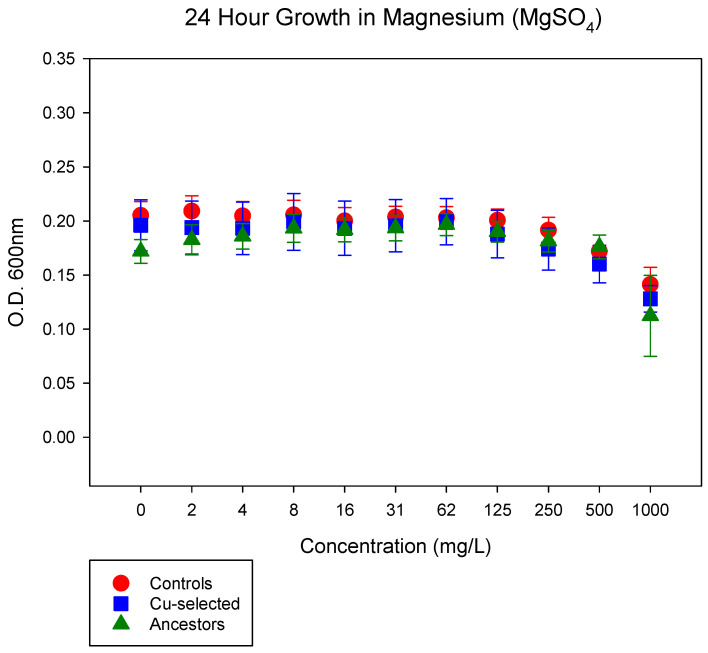
The mean and SE of 24 h growth for Cu-selected, control, and ancestor populations in increasing concentrations of MgSO_4_ are shown. There is no significant difference between the Cu-selected replicates populations when compared to the control and ancestor populations.

**Figure 3 antibiotics-11-00711-f003:**
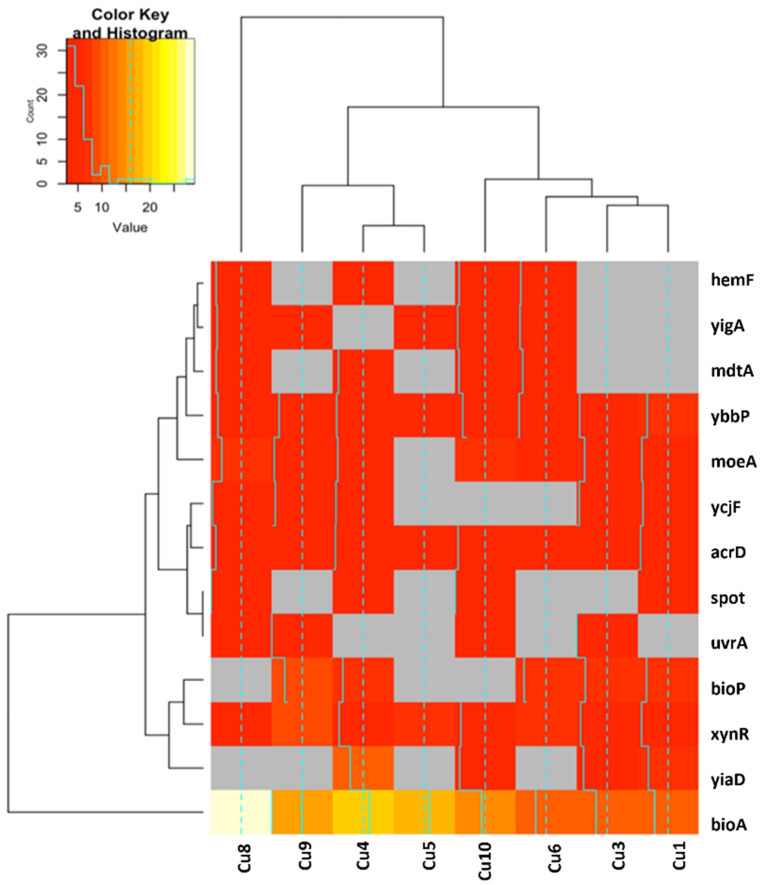
Hierarchical clustering of upregulated genes. Shown here are genes that were upregulated in at least four Cu-selected replicates with a 2-fold-change or above that were not expressed in the control populations. The rows and columns of the heatmaps are clustered based on similar observations.

**Figure 4 antibiotics-11-00711-f004:**
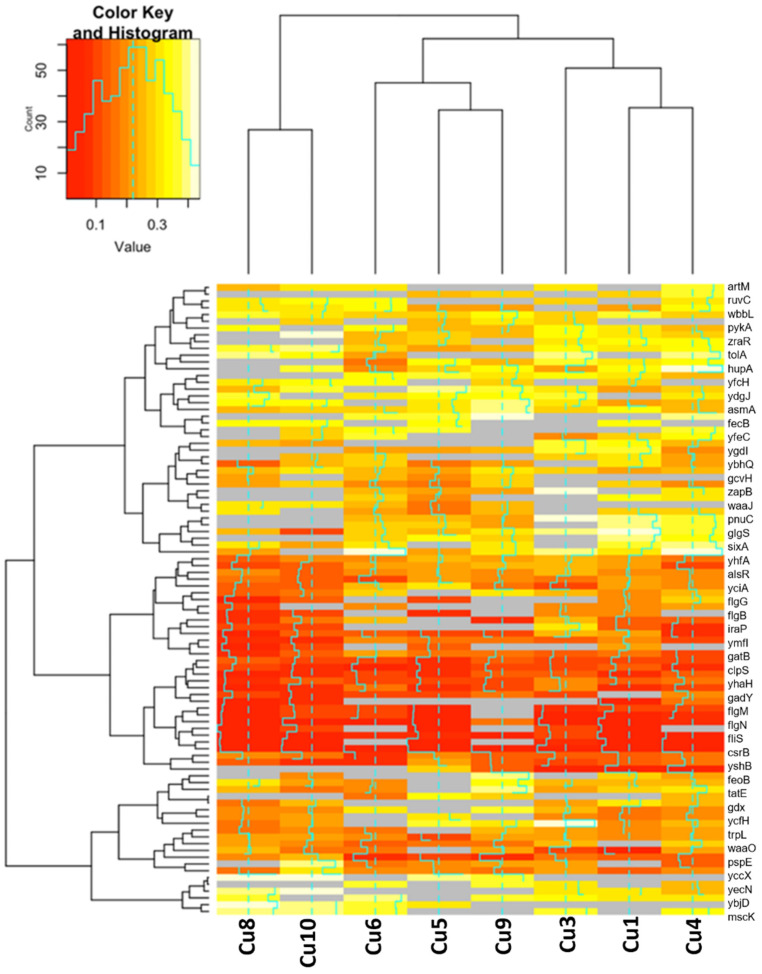
Hierarchical clustering of downregulated genes. Shown here are genes that were downregulated in at least four Cu-selected replicates with a 0.5-fold-change or below that were not expressed in the control populations. The rows and columns of the heatmaps are clustered based on similar observations.

**Figure 5 antibiotics-11-00711-f005:**
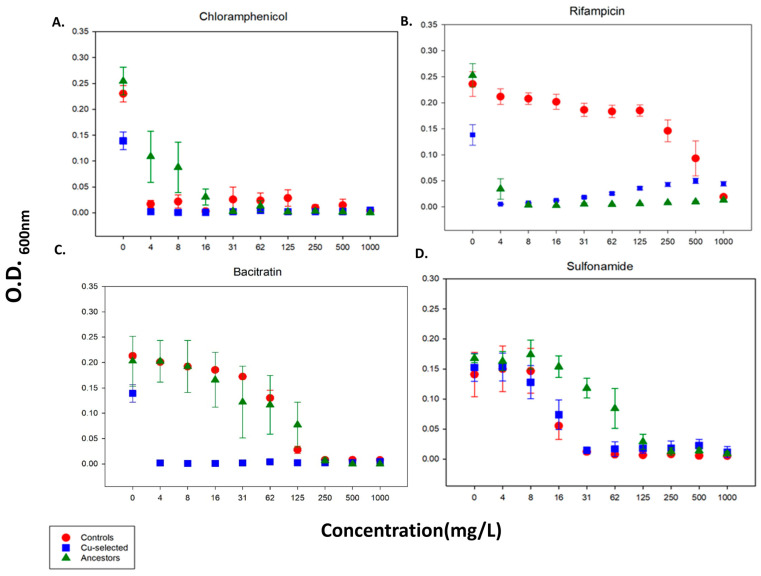
The mean and SE of 24 h growth for Cu-selected, control, and ancestor populations in increasing concentrations of the antibiotics chloramphenicol, bacitracin, rifampicin, and sulfonamide. The ancestor populations grew better than the Cu-selected and control populations in the presence of chloramphenicol (4–16 mg/L) (**A**); the control populations showed better growth than the ancestor and Cu-selected populations in rifampicin (**B**). In acitracin, the ancestor and control populations grew better than the Cu-selected populations at 4–125 mg/L (**C**). The ancestor populations also showed superior growth when compared to the Cu-selected populations in the presence of sulfonamide (16–62 mg/L) (**D**).

**Figure 6 antibiotics-11-00711-f006:**
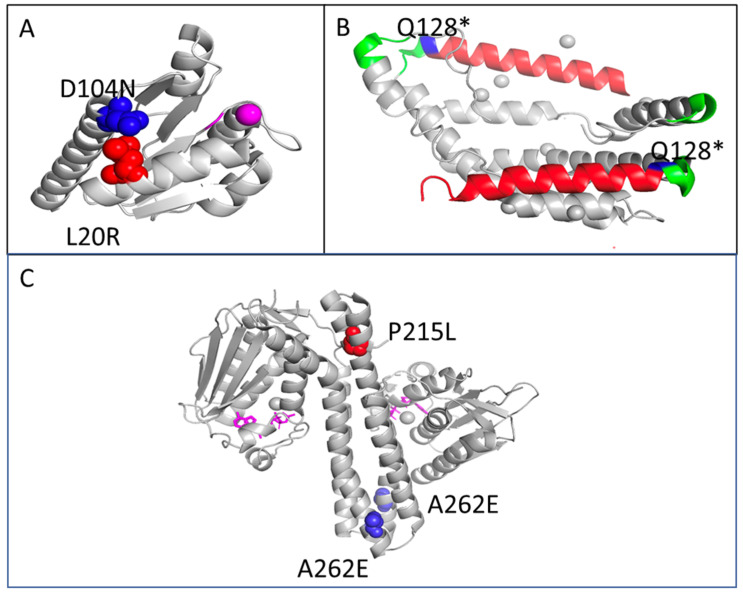
Mapping Cpx mutations onto three-dimensional protein structures. (**A**) CpxR mutations were mapped onto the solved structure of the receiver domain (PDB 4UHK); phosphorylated aspartate and magnesium ion are colored in magenta. All detected mutations were mapped, including D104N (blue) and L20R (red). (**B**) Q128* was the only detected CpxP mutation and was mapped onto the dimeric structure (PDB 3QZ3). Q128 resides in one of the two conserved LTXXQ motifs in each monomer (green), and this premature stop codon would halt translation of the final helix (red). (**C**) Currently, a crystal structure is only available for the cytoplasmic ATPase domain of CpxA (PDB 5LFK) and, therefore, only two mutations P215L (red and only structured in one monomer) and A262E (blue, present in both monomers) could be mapped.

**Table 1 antibiotics-11-00711-t001:** General linear model (GLM) results for phenotypic assays comparison for each population (Cu-selected, control, and ancestor) in each tested antimicrobial. The general linear model is an analysis of variance to determine the effect of the population (Cu-selected, control, and ancestral), the concentration (range tested for each antimicrobial), and their interaction. The GLM computes an F-statistic and the probability of achieving that F-statistic by chance (*p* value). NS means not significant.

Antimicrobial	Concentration Effect	Population Effect	Interaction (Concentration × Population Effect)
Cu-selected > Controls			
Copper (CuSO_4_)	F = 7.2 *p* < 0.001	F = 20.2 *p* < 0.001	NS
Chloramphenicol	NS	NS	NS
Sulfonamide	F = 16.5 *p* < 0.001	NS	NS
Cu-selected > Ancestor			
Copper (CuSO_4_)	F = 6.96 *p* < 0.001	F = 20.5 *p* < 0.001	NS
Rifampicin	NS	F = 21 *p* < 0.001	NS
Controls > Ancestors			
Copper (CuSO_4_)	F = 5.52 *p* < 0.001	NS	NS
Rifampicin	NS	F = 54 *p* < 0.001	F = 3 *p* < 0.05
Controls > Cu-selected			
Chloramphenicol	F = 23.5 *p* < 0.001	19.57 *p* < 0.001	NS
Bacitracin	F = 21.6 *p* < 0.001	NS	F = 2.14 *p* < 0.05
Magnesium	NS	NS	NS
Ancestor > Control			
Chloramphenicol	F = 4.1 *p* < 0.001	3.23 *p* < 0.005	NS
Bacitracin	F = 9.56 *p* < 0.001	NS	NS
Sulfonamide	F = 23.5 *p* < 0.001	F = 19 *p* < 0.001	NS
Ancestor > Cu-selected			
Bacitracin	F = 17 *p* < 0.001	F = 13 *p* < 0.001	NS
Rifampicin	F = 16 *p* < 0.001	F = 9.1 *p* < 0.05	F = 5 *p* < 0.001
Chloramphenicol	F = 4 *p* < 0.001	NS	F = 3 *p* < 0.05
Sulfonamide	F = 17.4 *p* < 0.001	F = 8.5 *p* < 0.05	NS

**Table 2 antibiotics-11-00711-t002:** Whole-genome resequencing results after 37 days of selection to CuSO_4_. Variant Color coding: brown, hard-selective sweep/fixation; green, hard selection, major variant; blue, minor variant.

		Cu-Selected Replicate Populations after 37 Days of Selection to CuSO_4_									
Gene Name	Mutations	Cu1	Cu2	Cu3	Cu4	Cu5	Cu6	Cu8	Cu9	Cu10	Gene Descriptions *Breseq and Uniprot
*cpxA*	K194T (AAA → ACA)	0.16	0.00	0.00	0.00	0.00	0.00	0.00	0.00	0.00	sensory histidine kinase in two-component regulatory system with CpxR
*cpxA*	K121R (AAA → AGA)	0.44	0.00	0.00	0.00	0.00	0.00	0.00	0.00	0.00	“
*cpxA*	V94A (GTG → GCG)	0.09	0.00	0.00	0.00	0.00	0.00	0.00	0.00	0.00	“
*cpxA*	R93H (CGC → CAC)	0.13	0.00	0.00	0.00	0.00	0.00	0.00	0.00	0.00	“
*cpxA*	K188Q (AAA → CAA)	0.00	1.00	0.00	0.00	0.00	0.00	0.00	0.00	0.00	“
*cpxA*	L178R (CTG → CGG)	0.00	0.00	0.35	0.00	0.00	0.00	0.00	0.00	0.00	sensory histidine kinase in two-component regulatory system with CpxR
*cpxA*	A78V (GCG → GTG)	0.00	0.00	0.00	1.00	0.00	0.00	0.00	0.00	0.00	“
*cpxR*	L20R (CTG → CGG)	0.00	1.00	0.00	0.00	0.00	0.00	0.00	0.00	0.00	response regulator in two-component regulatory system with CpxA
*cpxR*	D104N (GAT → AAT)	0.00	0.00	0.00	0.00	0.23	0.00	0.00	0.00	0.00	“
*cpxP*	F62L (TTT → CTT)	0.00	0.00	0.00	0.00	0.00	0.00	0.934	0.00	0.00	inhibitor of the cpx response; periplasmic adaptor protein
*cpxP*	Q128* (CAG → TAG)	0.00	0.00	0.00	0.00	0.00	0.00	0.00	0.00	0.87	“
*adhE*	coding (228/2676 nt)	0.00	0.00	0.00	0.00	0.385	0.00	0.00	0.00	0.00	fused acetaldehyde-CoA dehydrogenase/iron-dependent alcohol dehydrogenase/pyruvate-formate lyase deactivase
*adhE*	coding (143-147/2676 nt)	0.00	0.00	0.736	0.00	0.00	0.00	0.00	0.00	0.00	fused acetaldehyde-CoA dehydrogenase/iron-dependent alcohol dehydrogenase/pyruvate-formate lyase deactivase
*adhE*	V130I (GTT → ATT)	0.00	0.00	0.00	0.00	0.392	0.00	0.00	0.00	0.00	“
*rho*	coding (229/2676 nt)	0.00	0.00	0.00	0.00	0.41	0.00	0.00	0.00	0.00	transcription termination factor
*rho*	Q128* (CAG → TAG)	0.00	0.00	0.00	0.00	0.00	1.00	0.00	0.00	0.00	transcription termination factor
*rho*	Q128* (CAG → TAG)	0.00	0.00	0.00	0.00	0.00	0.00	0.00	0.81	0.00	transcription termination factor
*rpoD*	H600Q (CAC → CAG)	0.00	0.00	0.00	0.00	0.00	0.34	0.00	0.00	0.00	RNA polymerase, sigma 70 (sigma D) factor
*rpoD*	R109C (CGC → TGC)	0.00	0.00	0.00	0.00	0.00	0.50	0.00	0.00	0.00	“
*yddA*	P215L (CCA → CTA)	0.00	0.00	0.674	0.00	0.00	0.00	0.00	0.00	0.00	Inner membrane ABC transporter ATP-binding protein
*fis*	Δ1 bp	0.00	0.00	0.653	0.00	0.00	0.00	0.00	0.00	0.00	DNA-binding protein
*yidL*	coding (313-316/894 nt)	0.00	0.00	0.00	1.00	0.00	0.00	0.00	0.00	0.00	uncharacterized HTH-type transcriptional regulator
*yfcZ* /*fadL*	A262E (GCG → GAG)	0.00	0.00	0.00	1.00	0.00	0.00	0.00	0.00	0.00	family protein/long-chain fatty acid outer membrane transporter

“ means same gene definitions as above.

**Table 3 antibiotics-11-00711-t003:** Functional annotation clustering for upregulated and downregulated genes. There was one significant cluster for upregulated genes, which was associated with ABC binding. There were three significant clusters associated with bacterial flagellum, iron transport, and cell division among downregulated genes.

	Functional Annotation Cluster	Enrichment Score ^a^	Gene Count
Upregulated	Cluster 1 ABC transporter ATP-binding Nucleotide-binding ATP binding transport	1.34	17
Downregulated	Cluster 1 Bacterial flagellum	3.51	14
	Cluster 2 Iron transport Ion transport	1.37	11
	Cluster 3 Cell division Cell cycle	1.07	6

a. Enrichment score threshold was set to 1 and gene count ≥ 4.

## Data Availability

The datasets generated and analyzed in this study can be found through the NCBI BioProject database (https://www.ncbi.nlm.nih.gov/bioproject/) (accessed on 16 May 2021) under accession PRJNA730301.

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
