# Peer review of "Experimental Evolution of Copper Resistance in Escherichia coli Produces Evolutionary Trade-Offs in the Antibiotics Chloramphenicol, Bacitracin, and Sulfonamide"

_antibiotics, 2022, doi:10.3390/antibiotics11060711_

Round 1

Reviewer 1 Report

The manuscript by Sada Boyd et al. presents an evaluation of the experimental evolution of copper resistance in E. coli. This is of high importance in the field as the application of antimicrobial metals including copper may have an impact on antibiotic resistance and therefore therapeutic efficacy of antimicrobial treatments.  The authors performed in vitro evolution identified several mutations in Cu-adapted cells and analyzed the transcriptional changes in response to Cu. The Cu-adapted cells were tested for antibiotic resistance and showed higher sensitivity to bacitracin. Below are several comments and questions.

It is interesting whether the resistance to Cu was developing gradually. It would be insightful to see the growth data with Cu at earlier time points as well. Unless the population numbers reflect the time points, this needs to be clearly explained. Based on the provided data, it is difficult to see if the MIC for Cu is changed in the adapted cells. What is the MIC for Cu in the strain?

Fig 1 shows that the initial and the Cu-adapted cultures have drastic differences in growth at 0 Cu. However, in Mg experiment, there is no difference at 0. On the other hand, all the cultures in Mg experiment (0 mg/L) showed higher growth. I assume these all are the same cultures and the same growth conditions (at 0). Also, explain your controls in the results/figure legends. If the controls are the cultures that were passed at no Cu present, then why would they show increased resistance to Cu? Adaptation to the medium alone does not explain it as you do not see any difference at 0 Mg. Were the cultures tested for Mg stored for longer and therefore lost their growth advantage? By the way, does the DMB contain any Mg?

Table 1 needs more explanation in the legend. What do the numbers actually mean?

Table 2, if the populations are biological replicates, I would recommend to find a way to show the data in a more concise way.

It would help if the differentially regulated genes are organized into functionally meaningful categories. Efflux pumps are usually co-expressed multi-component systems. Did you see changes in the expression for only single components? The heat maps show some clustering, which needs to be explained.

Why not present the data for antibiotic resistance as MIC? Why would the control cells show such a difference in resistance to rifampicin? Again, adaptation to the medium does not explain this.

If CpxAR is important for Cu resistance, it is not clear how the detected mutations in cpxAR would be beneficial for Cu adaptation. On the other hand, if the TCS is mutated, it would impact the expression of its regulon. Did you see any changes in the transcription of the CpxR regulon?

The authors explain the mutations in rpo genes by adaptation to the medium, but in this case, all the cultures should have these mutations and not only Cu-adapted.

The authors relate the downregulation of flagellum biosynthesis to biofilm formation. Have you observed any cell aggregation?

LPS biosynthesis relies on a large number of genes. What are the functions of waaOL that could contribute to the changes in LPS structure?

It would help if at least some transcriptional changes were validated to play role in Cu resistance. This would strengthen the discussion points and conclusions.

Abstract, instead of “worse”, use more characteristic parameters.

Conclusions, the first two statements are not conclusions.

Other comments

  1. Provide a better explanation for the statistics in Table 1. It is difficult to follow. I would add the growth statistics to the appropriate graphs/legends.
  2. Spell out SE when mentioned the first time.
  3. L. 185-186 Edit the sentence
  4. L. 192 flagella biosynthesis?
  5. L. 516 “in at least four”
  6. L. 521 “to other”
  7. L. 283 It reads as acrD was shown to contribute to Cu resistance, but the reference only covers mdtD.
  8. L. 383 were
  9. L. 474-476 Edit for clarity
  10. L.484 Confusing about the group 4
  11. Table A4 is not clear what it shows.
  12. L.506 Edit for clarity

Author Response

Point 1: It is interesting whether the resistance to Cu was developing gradually. It would be insightful to see the growth data with Cu at earlier time points as well. Unless the population numbers reflect the time points, this needs to be clearly explained. Based on the provided data, it is difficult to see if the MIC for Cu is changed in the adapted cells. What is the MIC for Cu in the strain?

Response 1: We have been studying the evolution of metal resistance in this strain of E. coli since 2014. These studies have examined resistance to ionic and nanoparticle silver and iron, as well as ionic copper and gallium (see review in Graves JL 2020). In these studies, we demonstrated that resistance was associated with the appearance of specific mutations followed by hard selective sweeps of those variants in the metal-exposed treatments. For example, in Tajkarimi et al. 2017 we showed that silver resistance increased along the course of the experiment eight-fold from generation 60—200. This general phenomena in microbial evolution are well known for a variety of phenotypes (e.g., the Lenski LTEE for DMB medium, antibiotics, and temperature).  It is also known that the genomic basis of adaptation in bacteria changes as clonal interference replaces specific variants (e.g., Graves et al. 2015; Maddamsetti, Lenski, and Barrick 2015). For this reason, we stopped assaying our experiments at early points in the course of selection.  This is because the resistance phenotype and its genomics foundations at any given point of selection differs.

References

Graves, Ewunkem AJ, Thomas MD, Han J, Rhinehardt KL, Boyd S, Edmondson R, Jeffers-Francis L, and Harrison SH. 2020. Experimental Evolution of Metal Resistance in Bacteria. In Banzhaf et al. (Eds.) Evolution in Action—Past, Present, and Future, pp. 91—106, (Cham, Switzerland: Springer 2 International Publishing, AG).

Graves, J.L., Tajkarimi, M., Cunningham, Q., Campbell, A., Nonga, H., Harrison, S.H., Barrick, J.G., Rapid evolution of silver nanoparticle resistance in Escherichia coli, Frontiers in Genetics 6(42), 2015, doi: 10.3389/fgene2015.0042.

Maddamsetti R, Lenski RE, Barrick JE. Adaptation, Clonal Interference, and Frequency-Dependent Interactions in a Long-Term Evolution Experiment with Escherichia coli. Genetics. 2015; 200(2):619-31. doi: 10.1534/genetics.115.176677.

Tajkarimi M. Rhinehardt K. Thomas M. Ewunkem JA. Campbell A. Boyd S. Turner D. Harrison SH. and Graves JL. 2017. Selection for ionic- confers silver nanoparticle resistance in Escherichia coli. JSM Nanotechnology and Nanomedicine 5: 1047, 2017.

Point 2: Fig 1 shows that the initial and the Cu-adapted cultures have drastic differences in growth at 0 Cu. However, in Mg experiment, there is no difference at 0. On the other hand, all the cultures in Mg experiment (0 mg/L) showed higher growth. I assume these all are the same cultures and the same growth conditions (at 0). Also, explain your controls in the results/figure legends. If the controls are the cultures that were passed at no Cu present, then why would they show increased resistance to Cu? Adaptation to the medium alone does not explain it as you do not see any difference at 0 Mg. Were the cultures tested for Mg stored for longer and therefore lost their growth advantage? By the way, does the DMB contain any Mg?

Response 2: Figure 1 shows at 0 mg of additional copper (the condition of standard DMB medium) the mean and standard error for the 10 copper-adapted replicate populations, in comparison to the 5 control, and 5 ancestral replicates.  This figure indicates that there was not much within population variation in the 24-hr growth for any of these groups as the standard errors are very small.  Figure 3 shows the results from these same populations in magnesium sulfate (MgSO4.) Once again, the standard errors for all groups shown is small, indicating there was not much within selection treatment for growth in MgSO4.  The point of this comparison is to demonstrate that there was no apparent adaptation to increased amounts of the negative ion (SO4) in this study.  In other studies, such as Thomas et al. 2021, we did observe adaptation to increased amounts of the sulfate ion.

Reference

Thomas MD. Ewunkem J. Williams DK, Moore A. Rhinehardt K. Van Beveren E. Yang B. Boyd S. Tapia A. Han J. and Harrison SH. Graves JL, Too much of a good thing: Adaptation to iron (II) intoxication in Escherichia coli, Evolution, Medicine, and Public Health, 2021 (Finalist, George C. Williams Award, for best paper published in Evolution, Medicine, & Public Health, 2021).

Point 3: Table 1 needs more explanation in the legend. What do the numbers actually mean?

Response 3: This table reports the results of General linear model results for phenotypic assays comparison for each population (Cu-selected, Control, Ancestral) in each tested antimicrobial.  We will edit the table legend to say: General linear model (GLM) results for phenotypic assays comparison for each population (Cu-selected, Control, Ancestral) in each tested antimicrobial. The general linear model is an Analysis of variance to determine the effect of the population (Cu-selected, Control, Ancestral), the concentration (range tested for each antimicrobial), and their interaction. The GLM computes a F-statistic and the probability of achieving that F statistic by chance (p value).

Point 4: Table 2, if the populations are biological replicates, I would recommend finding a way to show the data in a more concise way.

Response 4: Our design is standard for studies in experimental evolution.  Replicate populations are established from a common ancestor (E. coli K-12 MG1655) and each is allowed to evolve along its own trajectory.  As mutations occur at random, the appearance of the same mutation in replicate populations is taken as strong evidence for the significance of that given mutation, as in the case of R15L in the cusS gene from our 2017 study of silver resistance.  We have reported our genomic sequencing results from these studies using this format for the last 8 years. Our papers have been accepted, with this table format in journals such as Evolution, Medicine, and Public Health (EMPH, impact factor 5.425, listed as #8 in 50 journals dealing with evolution; and Frontiers in Genetics, impact factor 3.789, #4 in open access genetics journals).  We do not think that an edit to this table is necessary.

Reference

Tajkarimi M. Rhinehardt K. Thomas M. Ewunkem JA. Campbell A. Boyd S. Turner D. Harrison SH. and Graves JL. 2017. Selection for ionic- confers silver nanoparticle resistance in Escherichia coli. JSM Nanotechnology and Nanomedicine 5: 1047, 2017.

Point 5: It would help if the differentially regulated genes were organized into functionally meaningful categories. Efflux pumps are usually co-expressed multi-component systems. Did you see changes in the expression for only single components? The heat maps show some clustering, which needs to be explained.

Response 5: We have organized the results of our differentially regulated genes into functionally meaningful categories originally found in the Appendix Table A3, we have moved this table up in the MS. It is now Table 3.

 Point 6: Why not present the data for antibiotic resistance as MIC? Why would the control cells show such a difference in resistance to rifampicin? Again, adaptation to the medium does not explain this.

Response 6: We have shown our phenotypic data over the last 8 years as 24-hour growth against increasing concentration.  There are alternative definitions of MIC, so our method avoids that controversy.  There is a strong body of literature that shows that adaptation to DMB medium via mutations in rpoB does account for rifampicin resistance (Conrad et al. 2010; Borrell et al. 2013; Alifano et al. 2015; Graves et al. 2015; Zaw, Emran, Lin 2018; Cutugno 2020; Thomas et al. 2021). Indeed, a rpoB mutation that we recovered in E. coli from Graves et al. 2015 (H526Y) was first discovered Mycobacterium smegmatis (Graves 2021).

References

Conrad TM, Frazier M, Joyce AR, Cho BK, Knight EM, Lewis NE, Landick R, Palsson BØ. RNA polymerase mutants found through adaptive evolution reprogram Escherichia coli for optimal growth in minimal media. Proc Natl Acad Sci U S A. 2010 Nov 23; 107(47):20500-5. doi: 10.1073/pnas.0911253107.

Borrell S, Teo Y, Giardina F, Streicher EM, Klopper M, Feldmann J, Müller B, Victor TC, Gagneux S. Epistasis between antibiotic resistance mutations drives the evolution of extensively drug-resistant tuberculosis. Evol Med Public Health. 2013; 2013(1):65-74. doi: 10.1093/emph/eot003.

Zaw MT, Emran NA, Lin Z. Mutations inside rifampicin-resistance determining region of rpoB gene associated with rifampicin-resistance in Mycobacterium tuberculosis. J Infect Public Health. 2018 Sep-Oct;11(5):605-610. doi: 10.1016/j.jiph.2018.04.005.

Alifano P, Palumbo C, Pasanisi D, Talà A. Rifampicin-resistance, rpoB polymorphism and RNA polymerase genetic engineering. J Biotechnol. 2015 May 20;202:60-77. doi: 10.1016/j.jbiotec.2014.11.024.

Cutugno L, Mc Cafferty J, Pané-Farré J, O'Byrne C, Boyd A. rpoB mutations conferring rifampicin-resistance affect growth, stress response and motility in Vibrio vulnificus. Microbiology (Reading). 2020 Dec; 166(12):1160-1170. doi: 10.1099/mic.0.000991.

Thomas MD. Ewunkem J. Williams DK, Moore A. Rhinehardt K. Van Beveren E. Yang B. Boyd S. Tapia A. Han J. and Harrison SH. Graves JL, Too much of a good thing: Adaptation to iron (II) intoxication in Escherichia coli, Evolution, Medicine, and Public Health, 2021 (Finalist, George C. Williams Award, for best paper published in Evolution, Medicine, & Public Health, 2021).

Graves JL. Principles and Applications of Antimicrobial Nanomaterials (Amsterdam: Elsevier), 2021.

Point 7: If CpxAR is important for Cu resistance, it is not clear how the detected mutations in cpxAR would be beneficial for Cu adaptation. On the other hand, if the TCS is mutated, it would impact the expression of its regulon. Did you see any changes in the transcription of the CpxR regulon?

Response 7: Previous studies have shown that CpxAR is activated in response to Cu toxicity line 260. Additionally, it has been shown that adaptation to Cu stress in E. coli and S. enterica requires activation of the envelope stress response regulators [Vogt et al., 2014; Pezza et al., 2016], which is monitored by CpxAR [Pezza et al., 2016; Raivio et al., 2013]. We suggest that cpxA mutations are important for Cu adaptation caused by envelope stress stimulated by excess Cu [Giachino et al., 2020]. This is supported by the fact that these mutations were only observed in the Cu-selected populations. We didn’t see any transcriptional changes in the CpxR regulon, but we did observe mutations in cpxR. We plan to investigate how these specific mutations impact Cu resistance in a future study.

References

Vogt SL, Evans AD, Guest RL et al. The Cpx envelope stress response regulates and is regulated by small noncoding RNAs. J Bacteriol 2014;196:4229–38. doi:10.1128/JB.02138-14

Pezza A, Pontel LB, López C et al. Compartment and signal-specific codependence in the transcriptional control of Salmonella periplasmic copper homeostasis. Proc Natl Acad Sci - PNAS 2016;113:11573–8. doi:10.1073/pnas.1603192113

Raivio TL, Price NL. The Escherichia coli Cpx Envelope Stress Response Regulates Genes of Diverse Function That Impact Antibiotic Resistance and Membrane Integrity. J Bacteriol 2013;195:2755–67. doi:10.1128/JB.00105-13

Giachino A, Waldron KJ. Copper tolerance in bacteria requires the activation of multiple accessory pathways. Mol Microbiol 2020.doi:10.1111/mmi.14522

Point 8: The authors explain the mutations in rpo genes by adaptation to the medium, but in this case, all the cultures should have these mutations and not only Cu-adapted.

Response 8: This claim is not true.  We have examined anti-metal adaptation for the last 8 years (silver, copper, iron, gallium).  Mutations in the rpo’s are common, but they are not always found. This indicates that natural selection has the capacity to find alternative mechanisms for adaptation to toxic metals.  The rpo mutations have often appeared in the controls due to adaptation to low sugar (Conrad 2010), but some also seem to have anti-metal properties.  This is supported by our molecular modeling studies associated with specific rpo mutations (see Graves 2021).

References

Conrad TM, Frazier M, Joyce AR, Cho BK, Knight EM, Lewis NE, Landick R, Palsson BØ. RNA polymerase mutants found through adaptive evolution reprogram Escherichia coli for optimal growth in minimal media. Proc Natl Acad Sci U S A. 2010 Nov 23; 107(47):20500-5. doi: 10.1073/pnas.0911253107.

Graves, Ewunkem AJ, Thomas MD, Han J, Rhinehardt KL, Boyd S, Edmondson R, Jeffers-Francis L, and Harrison SH. 2020. Experimental Evolution of Metal Resistance in Bacteria. In Banzhaf et al. (Eds.) Evolution in Action—Past, Present, and Future, pp. 91—106, (Cham, Switzerland: Springer 2 International Publishing, AG).

Graves JL. Principles and Applications of Antimicrobial Nanomaterials (Amsterdam: Elsevier), 2021.

Point 9: The authors relate the downregulation of flagellum biosynthesis to biofilm formation. Have you observed any cell aggregation?

Response 9: No, we did not observe cell aggregate previous studies suggest that the first step in biofilm formation is downregulation of flagellum, however we did not specifically investigate our populations for cell aggregation.

Point 10: LPS biosynthesis relies on a large number of genes. What are the functions of waaOL that could contribute to the changes in LPS structure?

Response 10: WaaL(rfaL) is involved in the LPS core biosynthesis pathway, which is part of the bacterial outer membrane biogenesis (UniProtKB - P27243). This protein adds the O-antigen on the glucose group of LPS. The O antigen provides protection from the antimicrobial action of certain antibiotics. WaaO (rfaI) is an LPS 1,3 galactosyltranferase involved the synthesis of the R core lipopolysaccharide (UniProtKB - Q9ZIS5). The function of the LPS is to provide structural integrity and to create a permeability barrier that protects bacteria from toxins. Perhaps downregulating genes involved in LPS biosynthesis is a mechanism of resistance against Cu. We have included this information in our MS for clarification.

Point 11: It would help if at least some transcriptional changes were validated to play role in Cu resistance. This would strengthen the discussion points and conclusions.

Response 11: The evidence for the fact that these transcriptional changes do play a role in copper resistance is the fact that they appear in the copper-selected populations and do not appear in their controls. This is how experimental evolution with resequencing works as a method to uncover adaptation.  As the molecular mechanisms of adaptation are often complex, we have found that sometimes the mutations occur in loci that have a clear and direct relationship to the phenotype, e.g. in Thomas et al. 2021 we found mutations in several genes directly associated with iron metabolism (cueR, fecA), as well as substantial changes in transcription in those genes.  However, in that study we also observed selective sweeps in genes whose relationship to iron metabolism was not clear at all (murC, mrdA, yeaG, ptsP, yidX, rpoB; nor were their high frequencies simply the result of linkage).  Thus, what experimental evolution with resequencing provides us are resistant phenotypes and their genomic foundations.  The method often uncovers mechanisms were not found by classical mutational (loss of function) type studies.  As we have limited time and resources, we cannot investigate the molecular basis of every variant we find, instead these results provide the basis for further study concerning how these mechanisms may be associated with the resistance phenotype.

Point 12: Abstract, instead of “worse”, use more characteristic parameters.

Response 12: We have addressed this issue in our abstract.

Point 13: Conclusions, the first two statements are not conclusions.

Response 13: We deleted the first two statements of the discussion sections.

Points 14-25: Other comments

  1. Provide a better explanation for the statistics in Table 1. It is difficult to follow. I would add the growth statistics to the appropriate graphs/legends.
  2. 185-186 Edit the sentence
  1. L. 192 flagella biosynthesis? legends.
  2. Spell out SE when mentioned the first time.
  3. L. 516 “in at least four” legends.
  4. L. 521 “to other”
  5. L. 283 It reads as acrD was shown to contribute to Cu resistance, but the reference only covers mdtD.

Taken from Nishino et al., 2007. “ In addition to their role in multidrug resistance, we found that BaeSR, AcrD, and MdtABC contribute to copper and zinc resistance in Salmonella”.

  1. L. 383 were
  2. L. 474-476 Edit for clarity
  3. L.484 Confusing about the group 4
  4. Table A4 is not clear what it shows.
  5. L.506 Edit for clarity

Response 14-25: We have addressed the additional comments above.

Reviewer 2 Report

The manuscript by Sada M. Boyd et al. describes interesting data on the impact of copper exposures on the development of copper-resistant bacterial strains and their AST profile.

The manuscript is very interesting but deserves to be completed before possible publication.

Global:

use the passive voice

italicize all bacterial names (salmonella)

numbers less than twelve should be written in capital letters.

Results: consider the determination of the threshold in the methods part of the manuscript and not the results (line 173 to 175 for example).

Results/Methods: a major limitation is that a full AST has not been determined to clearly understand the impact of copper on antibiotic resistance. Please consider a full AST as recommended by CLSI or EUCAST.

Discussion: authors should discuss the impact on infectious disease management. For example, although the concentration should not be similar, have diseases such as Wilson's disease been associated with modulation of AST of the bacteria considered for example?

Author Response

Point 1: Global:

use the passive voice

italicize all bacterial names (salmonella)

numbers less than twelve should be written in capital letters.

We are aware that numbers under ten should be spelled out however we are not familiar with capitalizing number under twelve

Results: consider the determination of the threshold in the methods part of the manuscript and not the results (line 173 to 175 for example).

Lines 173-175 explains the results from the methods described in 4.7.2.

Response 1: We have addressed the above comments in our manuscript.

Point 2: Results/Methods: a major limitation is that a full AST has not been determined to clearly understand the impact of copper on antibiotic resistance. Please consider a full AST as recommended by CLSI or EUCAST.

Response 2: This study is part of a series of studies in the experimental evolution of metal resistance (ionic, nanoparticle, silver, copper, iron, gallium).  These studies have established that metal adaptation results in pleiotropic impacts resulting in antibiotic resistance.  It was never our intention to do a full AST profile (and we are unlikely to do that, as that is not our primary research interest).  However, our results represent a “proof of principle” for any researchers who would be interested in conducting such work.

References

Graves, Ewunkem AJ, Thomas MD, Han J, Rhinehardt KL, Boyd S, Edmondson R, Jeffers-Francis L, and Harrison SH. 2020. Experimental Evolution of Metal Resistance in Bacteria. In Banzhaf et al. (Eds.) Evolution in Action—Past, Present, and Future, pp. 91—106, (Cham, Switzerland: Springer 2 International Publishing, AG).

Graves JL. Principles and Applications of Antimicrobial Nanomaterials (Amsterdam: Elsevier), 2021.

Point 3: Discussion: authors should discuss the impact on infectious disease management. For example, although the concentration should not be similar, have diseases such as Wilson's disease been associated with modulation of AST of the bacteria considered for example?

Response 3: We are happy to expand our discussion of the implications of antimicrobial pleiotropy on infectious disease, as we have in our other publications.

Round 2

Reviewer 1 Report

The manuscript has been edited and the authors have addressed some of the concerns.

However, several questions remained.

In figure 1, for example, the authors state that “Cu-selected replicate populations demonstrated superior growth in CuSO4 concentrations of 8 to 62 mg/L”. However, I see that those populations grow better even at no Cu. Besides, comparing Fig 1 and Fig 3 raises a question of why the ancestor populations show such different growth at 0, where the conditions should be same for Cu and Mg experiments as the same medium was used: less than 0.1 for 0 Cu and about 0.2 for 0 Mg. This needs to be at least acknowledged.

As the authors indicated, CpxAR is activated by Cu and regulates genes involved in cell envelope stress response. It would make sense if these genes are at least protected from mutations in Cu-resistant populations, unless the mutations lead to their enhanced functions, for example, increased sensitivity of the sensor component to Cu. The opposite is expected from the mutations in cpxP, inhibiting the CpxAR response. Would it be possible to predict whether the cpxP mutations are detrimental for cpxP function?  Mapping these mutations on the protein domains may be insightful.

Author Response

Point 1: In figure 1, for example, the authors state that “Cu-selected replicate populations demonstrated superior growth in CuSO4 concentrations of 8 to 62 mg/L”. However, I see that those populations grow better even at no Cu.

Response 1: We are aware that the Cu-selected populations also demonstrate superior growth in the presence of DMB only without the addition of CuSO4 . We have repeated this experiment multiple times and have consistently observed that the Cu-selected populations demonstrate superior growth in the varying concentrations of CuSO4 while growth in the no environment tends to vary among populations. We will carefully consider this important point in our future experiments. However, the primary objective of figure 1 was to observe the differences in growth in the presence of CuSO4 to determine whether the Cu-selected populations have a selective advantage in the presence of Cu. We have also addressed this in our discussion.

Point 2: Besides, comparing Fig 1 and Fig 3 raises a question of why the ancestor populations show such different growth at 0, where the conditions should be same for Cu and Mg experiments as the same medium was used: less than 0.1 for 0 Cu and about 0.2 for 0 Mg. This needs to be at least acknowledged.

Response 2: We realize the importance of Point 2 however; the point of this comparison was to demonstrate that there was no apparent adaptation to increased amounts of the negative ion (SO4) in this study.  We have consistently seen variation in growth in the DMB media only environment however our primary objective of this study was to observe the differences in growth in the metal.

Point 3: As the authors indicated, CpxAR is activated by Cu and regulates genes involved in cell envelope stress response. It would make sense if these genes were at least protected from mutations in Cu-resistant populations, unless the mutations lead to their enhanced functions, for example, increased sensitivity of the sensor component to Cu. The opposite is expected from the mutations in cpxP, inhibiting the CpxAR response. Would it be possible to predict whether the cpxP mutations are detrimental for cpxP function?  Mapping these mutations on the protein domains may be insightful.

Response 3: We have addressed the comments in our discussion section and have mapped the Cpx mutations onto a three-dimensional protein shown in Figure 6. The Q128* mutation detected in CpxP results in a premature stop codon that would prevent translation of an entire helix. CpxP (PDV 3QZ3) carries two conserved LTXXQ motifs (green) and Q128 (blue) is in the second of these conserved motifs. Based on this mutation leading to a premature stop codon it is very likely that this mutation disrupts the structure enough that it results in a non-functional protein.

Reviewer 2 Report

The manuscript has been revised according to my previous comments.

Author Response

Point 1: The manuscript has been revised according to my previous comments.

Response 1: We thank you for your thoughtful comments and suggestions.